# Gradient Boosting for the Spectral Super-Resolution of Ocean Color Sensor Data

**DOI:** 10.3390/s25206389

**Published:** 2025-10-16

**Authors:** Brittney Slocum, Jason Jolliff, Sherwin Ladner, Adam Lawson, Mark David Lewis, Sean McCarthy

**Affiliations:** U.S. Naval Research Laboratory, Stennis Space Center, Bay St. Louis, MS 39529, USA; jason.k.jolliff.civ@us.navy.mil (J.J.); sherwin.d.ladner.civ@us.navy.mil (S.L.); timothy.a.lawson24.civ@us.navy.mil (A.L.); mark.d.lewis54.civ@us.navy.mil (M.D.L.);

**Keywords:** remote sensing, hyperspectral, machine learning

## Abstract

We present a gradient boosting framework for reconstructing hyperspectral signatures in the visible spectrum (400–700 nm) of satellite-based ocean scenes from limited multispectral inputs. Hyperspectral data is composed of many, typically greater than 100, narrow wavelength bands across the electromagnetic spectrum. While hyperspectral data can offer reflectance values at every nanometer, multispectral sensors typically provide only 3 to 11 discrete bands, undersampling the visible color space. Our approach is applied to remote sensing reflectance ($R_{rs}$) measurements from a set of ocean color sensors, including Suomi-National Polar-orbiting Partnership (SNPP) Visible Infrared Imaging Radiometer Suite (VIIRS), the Ocean and Land Colour Instrument (OLCI), Hyperspectral Imager for the Coastal Ocean (HICO), and NASA’s Plankton, Aerosol, Cloud, Ocean Ecosystem Ocean Color Instrument (PACE OCI), as well as in situ $R_{rs}$ data from National Oceanic and Atmospheric Administration (NOAA) calibration and validation cruises. By leveraging these datasets, we demonstrate the feasibility of transforming low-spectral-resolution imagery into high-fidelity hyperspectral products. This capability is particularly valuable given the increasing availability of low-cost platforms equipped with RGB or multispectral imaging systems. Our results underscore the potential of hyperspectral enhancement for advancing ocean color monitoring and enabling broader access to high-resolution spectral data for scientific and environmental applications.

## 1. Introduction

Hyperspectral imaging (HSI) captures fine spectral resolution across hundreds of contiguous bands, enabling detailed environmental analysis based on spectral signatures [1]. Interest in hyperspectral data is growing with advances in in situ instruments (e.g., WETLabs ac-s) and satellite sensors such as NASA’s Plankton, Aerosol, Cloud, Ocean Ecosystem Ocean Color Instrument (PACE OCI), though pathways for utilizing these high-dimensional datasets are in their early stages. The WETLabs ac-s is a ship-based spectrophotometer that provides high-resolution in-water absorption and attenuation spectra, making it a valuable source of reference data, though its coverage is limited to discrete sampling locations [2]. PACE OCI, by contrast, is a spaceborne hyperspectral sensor that provides global observations of ocean color with high spectral resolution (5 nm across the UV to NIR range), enabling the detection of subtle biogeochemical signals [3]. However, like many spaceborne HSI systems, it faces trade-offs in spatial resolution (1 km at nadir), data volume, and latency, which may limit its utility in fine-scale or near-real-time applications. The high spectral fidelity of HSI is particularly valuable in ocean color applications, where subtle changes in water-leaving radiance reveal key biogeochemical and physical processes, but the low spatial resolution and high data cost can limit its practicality for large-scale or time-sensitive remote sensing.

In contrast, multispectral imagery (MSI), which captures fewer, broader bands, is more readily available and often deployed from satellites, UAVs, and low-cost airborne platforms. For example, San Francisco-based Planet has more than two hundred nanosatellites currently orbiting the earth capturing near-daily MSI at 3-m resolution using commercial-grade 4-band sensors (PlanetScope) and 8-band sensors (SuperDove). Despite their accessibility, these systems lack the spectral granularity of HSI, limiting their use in applications requiring detailed spectral information. This disparity has driven significant interest in spectral super-resolution (SSR)—the task of reconstructing high-resolution hyperspectral data from limited MSI input.

SSR research has increasingly employed machine learning and deep learning techniques to model nonlinear relationships between MSI and HSI data. These approaches range from traditional regression models to advanced network architectures like temporal fusion frameworks and unfolding-based designs that integrate physics-based priors with data-driven learning [4]. Notably, models like the Multitemporal Spectral Reconstruction Network (MTSRN) have demonstrated the value of leveraging temporal sequences to improve spectral reconstruction fidelity, particularly in satellite time series [5].

While deep learning methods such as CNNs, transformers, and other deep-learning architectures have achieved strong performance in spectral reconstruction, they often require large training datasets, significant computational resources, and lack interpretability. In contrast, tree-based ensemble models like gradient boosting offer greater transparency, lower data and computational requirements, and strong performance on tabular or structured data. This makes them attractive for applications with limited ground truth data or constrained deployment environments. A gradient boosting ensemble was chosen over a decision tree due to the gradient boosting algorithm’s robustness to noise, nonlinearity, memory efficiency, and quick processing of large datasets [6].

Building on this trajectory, our work investigates the application of machine learning to predict hyperspectral reflectance in the visible range (400–700 nm) from MSI observations. The focus is on remote sensing reflectance ($R_{rs}$), a key parameter in ocean color applications. Using data from ocean color sensors, hyperspectral in situ measurements, and synthetic hyperspectral data, we train a gradient boosting regressor to reconstruct hyperspectral $R_{rs}$ spectra from limited-band MSI input. This approach holds potential for enhancing MSI data collected by non-traditional platforms (e.g., UAVs, small satellites) to support coastal ocean monitoring, particularly where high spatial resolution is required, and traditional hyperspectral sensors fall short.

## 2. Materials and Methods

### 2.1. Convolutions

To train a model capable of reconstructing high-resolution hyperspectral signatures from multispectral data sources, paired datasets are required with multispectral data as inputs and corresponding hyperspectral data as targets. The datasets used for this work were derived from two hyperspectral sources covering the visible range. These sources include a global sampling of a single day’s worth of data from PACE OCI, which covers 400 to 720 nm at 1 nm spectral resolution, and a synthetic hyperspectral reference spectra dataset, which covers the range of 400 to 700 nm at 1 nm spectral resolution. The synthetic dataset was established by the International Ocean-Colour Coordinating Group [7]. It is designed to simulate the hyperspectral $R_{rs}$ for a wide range of different water turbidities using variable Inherent Optical Properties (IOPs) as input to the HYDROLIGHT numerical radiative transfer simulation [8,9]. The spectra are representative of a range of marine conditions from very clear open ocean environments to very turbid coastal scenes. Water turbidity in this context is not described by a single metric but is implicitly controlled by varying the Inherent Optical Properties (IOPs) of the water. These include absorption and scattering coefficients for pure water, colored dissolved organic matter (CDOM), phytoplankton, and non-algal particles, as well as the volume scattering function. By manipulating these parameters, the dataset spans a continuum of optical water types, ranging from clear oceanic waters to highly turbid coastal and estuarine environments. The HYDROLIGHT model simulates how light propagates through water bodies, accounting for absorption, scattering, and internal reflection processes to produce realistic above-water $R_{rs}$ by solving the radiative transfer equation under specified environmental and optical conditions, including the IOPs described above [10].

To ensure one-to-one alignment between inputs and hyperspectral targets while retaining the spectral characteristics of real multispectral sensors, we used the hyperspectral data as the ground truth and simulated multispectral data by convolving the hyperspectral signatures with each sensor’s Relative Spectral Response (RSR) function. The RSR function describes how sensitive a remote sensing sensor’s spectral band is to different wavelengths of light within that band. This function characterizes the spectral profile of a sensor’s detector system across a given wavelength range. As such, it is important to use each sensor’s RSR when creating the training data so that the algorithm can learn the response of that sensor. Since the PACE OCI data extends beyond 700 nm, prior to convolution, the hyperspectral data were trimmed to 400–700 nm to match the spectral range of the synthetic dataset. Missing and invalid values in the spectral bands were interpolated to achieve a uniform 1 nm spectral resolution. We also removed spectra affected by cloud cover, glint, stray light, or land contamination from the PACE OCI data to ensure data quality.

To simulate the spectral characteristics of the input multispectral sensors, we applied each sensor’s Relative Spectral Response (RSR) function to the original hyperspectral data. Each RSR function was used to convolve the full hyperspectral data by performing a weighted average over the relevant wavelengths as shown in Equation (Equation 1), where $R_{sensor,i}$ is the $R_{rs}$ for band *i* of the sensor of interest, $R_{HSI}\left(\lambda \right)$ is the $R_{rs}$ at wavelength λ in the original hyperspectral data, and $RSR_{i}\left(\lambda \right)$ is the spectral response of band i at wavelength λ. This convolution step results in a multispectral-like representation that matches the spectral characteristics of the target sensor. These convolved values serve as the input that the model will use to learn to recreate the full range of hyperspectral values.(1)$$R_{sensor,i}=\frac{\sum_{\lambda }R_{HSI}\left(\lambda \right)\cdot RSR_{i}\left(\lambda \right)}{\sum_{\lambda }RSR_{i}\left(\lambda \right)}$$

### 2.2. Colorimetry

Colorimetry is the science of numerically quantifying color based on the spectral power distribution, ϕ(λ), across the electromagnetic spectrum [11]. In this study, we apply colorimetric analysis to evaluate the spectral balance and coverage of our datasets within the visible range (400–700 nm). To do this, we rely on the CIE 1931 color space, developed by the International Commission on Illumination (CIE), which maps the visible spectrum to human color perception using standardized tristimulus functions [12,13]. These functions correspond to the human eye’s sensitivity to red, green, and blue light, respectively. While the foundational principles of colorimetry have remained largely unchanged, recent work has continued to refine color difference metrics and extend colorimetric analysis into modern domains such as hyperspectral imaging and remote sensing [14,15,16]. These advances support more perceptually accurate comparisons and improved visualization techniques, particularly when evaluating high-dimensional spectral data. A recent study highlighted the usefulness in using chromaticity indices over a unified spectral range to distinguish algal bloom waters from normal waters, underscoring the importance of knowing the chromaticity space of a remote-sensed dataset [17].

In their original form, Lee [7] defines the tristimulus values X, Y, and Z using the integrals presented in Equation (Equation 2). Here, F(λ) is the spectral power distribution, and $K_{m}$ = 638 lm $W^{-1}$ is the maximum luminous efficacy of radiant power under photopic conditions.(2)$$X=K_{m}\int_{0}^{\infty }F\left(\lambda \right)\bar{x}\left(\lambda \right)d\lambda ,\;Y=K_{m}\int_{0}^{\infty }F\left(\lambda \right)\bar{y}\left(\lambda \right)d\lambda ,\;Z=K_{m}\int_{0}^{\infty }F\left(\lambda \right)\bar{z}\left(\lambda \right)d\lambda$$

To adapt these definitions to ocean color remote sensing, we substitute F(λ) with $R_{rs}\left(\lambda \right)$ and apply sensor-specific relative spectral response (RSR) functions to simulate each channel’s sensitivity. The modified equations are given in Equation (Equation 3) where $SR_{R}$, $SR_{G}$, and $SR_{B}$ represent the RSRs of the red, green, and blue channels, respectively.(3)$$X=\int_{400}^{700}\pi R_{rs}\left(\lambda \right)SR_{R}\left(\lambda \right)d\lambda ,\;Y=\int_{400}^{700}\pi R_{rs}\left(\lambda \right)SR_{G}\left(\lambda \right)d\lambda ,\;Z=\int_{400}^{700}\pi R_{rs}\left(\lambda \right)SR_{B}\left(\lambda \right)d\lambda$$

The resulting X, Y, and Z values represent the red, green, and blue channel responses derived from the $R_{rs}$ spectrum of each pixel. These can be converted into chromaticity coordinates (x,y,z), which describe the color independently of brightness, as follows:(4)$$x=\frac{X}{X+Y+Z},\;y=\frac{Y}{X+Y+Z},\;z=\frac{Z}{X+Y+Z}$$

In practice, we visualize color coverage using only the 2D chromaticity space (x,y). This allows us to assess how well our datasets represent the range of ocean colors. Chromaticity plots were generated for the synthetic hyperspectral dataset, the PACE OCI training data, and their combined coverage (Figure 1). Plotting ocean color data in chromaticity space serves as a quality control tool, helping to identify underrepresented regions of the color spectrum. For example, the synthetic dataset includes a broader range of blue and green waters, while the PACE OCI dataset captures more mixed-color waters with red–green–blue blends. By combining both datasets, we achieve more comprehensive chromaticity coverage, improving the training base for color reconstruction models.

The globally collected PACE OCI dataset contains over 500 thousand valid pixels, so to reduce the size of the global dataset while preserving spectral variability, KMeans clustering was applied to subset the data based on clusters found in the distribution of hyperspectral bands. Specifically, we selected five representative wavelengths (400 nm, 450 nm, 500 nm, 550 nm, and 600 nm) to characterize the spectral diversity of each pixel. These wavelengths span the visible spectrum and capture key absorption and reflectance features relevant to ocean color products. Each pixel’s $R_{rs}$ values at these five wavelengths were used as a 5-dimensional feature vector, which the KMeans clustering algorithm used to partition the dataset into 6 clusters. The 6 clusters were selected to be representative of the six overall trends in the distribution of red, green, and blue bands seen when plotting the spectra for the training data. From each cluster, 5% of the representative samples were selected to form a spectrally diverse but compact training set. This approach ensured that the final training dataset retained a balanced representation of the spectral variability present in the original scene, while reducing the computational burden of training on the full dataset [18,19].

### 2.3. Model Development

To develop a model that reconstructs hyperspectral signatures from multispectral inputs, we developed a supervised regression model using a gradient boosting algorithm using Scikit Learn’s GradientBoostingRegressor. Gradient boosting is a machine learning technique used for regression problems that works by combining many decision trees into a strong predictive model. As such, the model is built sequentially such that each new tree is trained to correct the errors made by the previous ones. Because it can capture non-linear relationships and automatically handle feature interactions, gradient boosting is widely used in structured data problems [20]. The model created for this research consists of a GradientBoostingRegressor as the base estimator, encapsulated within a MultiOutputRegressor—both modules obtained from Scikit Learn’s ensemble package [21]. This setup trains a separate gradient boosting model for each target output (i.e., each wavelength), effectively addressing the high-dimensional regression task posed by hyperspectral reflectance reconstruction. All hyperparameters of the gradient boosting model followed Scikit-Learn’s defaults, with the exception of max_depth, which was increased from 3 to 5 to allow slightly more complex trees.

The spectrally downsampled dataset obtained through KMeans clustering was used to train a set of gradient boost models—one per supported sensor. Each model learned to predict the full hyperspectral spectrum (400–700 nm) from the corresponding sensor-convolved input spectrum. For each sensor, a separate model was trained using the sensor-specific convolved inputs as features and the full-resolution hyperspectral signatures as targets. This per-sensor modeling approach allowed each regressor to learn the optimal mapping from its unique spectral profile to the underlying hyperspectral information.

Using the approach defined above, the following models were developed with input features based on the real, central wavelengths and bandwidths for each ocean color sensor:5-band SNPP VIIRS (410, 443, 486, 551, 671)7-band MODIS AQUA (412, 443, 488, 531, 547, 667, 678)11-band OLCI 3A (400, 412, 443, 490, 510, 560, 620, 665, 674, 682, 709)3-band Planet PlanetScope (490, 560, 645)

Each model was trained using a train–test split allocating 80% of the data to training and the other 20% for validation. The models were trained on the synthetic data, the globally sampled PACE OCI data, and a combination of the two datasets to examine which distribution yields preferable performance. The models were trained and validated using remote-sensing $R_{rs}$ data from multiple sources prepared through the use of sensor convolutions and balanced downsampling as described above. For evaluation purposes, we assessed performance on hyperspectral $R_{rs}$ data collected from the Hyperspectral Imager for the Coastal Ocean (HICO), a Rhode Island scene with collocated PACE OCI and multispectral sensor data, and above-water spectroradiometer measurements obtained during the annual NOAA calibration and validation cruises from 2019 and 2022 [22,23]. The $R_{rs}$ for HICO were downloaded from the NASA Ocean Color website [24].

The Rhode Island multi- and hyper-spectral scenes were processed using NRL’s Automated Processing System (APS). APS is based on NASA’s SeaDAS codeset and used to ingest multi- and hyperspectral remote sensing data from continuously imaging ocean color sensors to automatically produce numerous products of interest to Navy operations and inputs to bio-optical forecasting models [25,26]. These Navy support products are derived from optical properties, including but not limited to vertical and horizontal visibility products produced from Apparent Optical Properties (AOPs), Inherent Optical Properties (IOPs), and Electro Optical system performance products for cameras, lasers, and divers.

## 3. Results

### 3.1. Training Time Performance

To evaluate each model’s performance across the 300 predicted wavelengths, we computed the average root mean squared error (RMSE), which summarizes the squared deviation of the prediction from the true $R_{rs}$ value across all wavelengths. Defining N as the number of spectra and M as the number of wavelengths from 400 nm to 700 nm for a 1 nm spectral resolution, the average RMSE is computed as shown in Equation (Equation 5), where Y is the true $R_{rs}$ value and $\hat{Y}$ is the predicted $R_{rs}$ value. Additionally, to account for variation in scale across spectral bands, we applied Min–Max normalization and computed the average coefficient of determination ($R^{2}$) on the scaled data. The equation for this process is given in Equation (Equation 6). This provides a normalized measure of how well the model captures spectral variability across the full $R_{rs}$ spectrum. As the models each output 300 values, 1 per nanometer, these RMSE and $R^{2}$ statistics reflect the average performance across all wavelengths.(5)$$RMSE=\sqrt{\frac{1}{NM}\sum_{i=1}^{N}\sum_{j=1}^{M}{\left(Y_{i,j}-{\hat{Y}}_{i,j}\right)}^{2}}$$(6)$$R^{2}=1-\frac{\sum_{i=1}^{N}\sum_{j=1}^{M}{\left(Y_{i,j}-{\hat{Y}}_{i,j}\right)}^{2}}{\sum_{i=1}^{N}\sum_{j=1}^{M}{\left(Y_{i,j}-{\bar{Y}}_{j}\right)}^{2}},\;where\;{\bar{Y}}_{j}=\frac{\sum_{i=1}^{N}Y_{i,j}}{N}$$

The values for each metric obtained at the time of training on the validation data are given in Table 1. As anticipated based on the chromaticity distributions, the models strengthened as the dataset was expanded, with the lowest performance on the synthetic dataset, increased performance on the global PACE OCI dataset, and most models seeing a further improvement when the two were used together. The MODIS 7-band model was an exception as the performance peaked with the use of the global PACE OCI data and saw a decrease upon adding the synthetic data to the training set.

Examination of the RMSE and $R^{2}$ per wavelength band shown in Figure 2 demonstrates that the error for the 3-band model (PlanetScope) greatly exceeds that of the other models when the wavelength is below 490 nm. This is expected as this model has no informative bands below that point. Aside from this, the 3-band model performs closely with the other models, meaning the majority of the decrease in the $R^{2}$ and increase in RMSE are attributed to these lower bands. All models experienced an uptick in error between 560 and 500 nm, which is the largest gap in bands in the multispectral sources, as well as near 690 nm. The 690 nm band is not far from nearby bands for many of the multispectral sensors, indicating that this difference is likely a consistent issue in the training data that the models have picked up.

### 3.2. Validation Performance

To evaluate model performance across multiple hyperspectral sources and varying conditions, we tested all four models on three independent validation datasets. The datasets consist of the NOAA calibration cruise data, three HICO scenes, and a scene over Rhode Island where MODIS, VIIRS, OLCI, and PACE OCI were available on the same day. For the HICO and in situ datasets, the data was convolved to each sensor’s RSR and those bands were used to predict the hyperspectral curve. For the Rhode Island scene, the raw input from each of the multispectral sensors for the Rhode island region was fed directly to the model and compared to the PACE OCI scene for that day.

The models were again assessed using measures of $R^{2}$ and average RMSE as well as Symmetric Mean Absolute Percentage Error (SMAPE). SMAPE is a metric that measures the relative accuracy of a model by comparing how far off predictions are from actual values as a percentage. The equation is given in Equation (Equation 7), where $y_{i}$ is the actual value, ${\hat{y}}_{i}$ is the predicted value, and *n* is the number of observations. Unlike standard MAPE, which divides by the actual value, SMAPE divides by the average of the actual and predicted values, making it more balanced when the true value is very close to 0. Since SMAPE is computed for all wavelengths at all points, this again represents the average performance across all bands.(7)$$SMAPE=\frac{100}{n}\sum_{i=1}^{n}\frac{|y_{i}-{\hat{y}}_{i}|}{\frac{\left(|y_{i}\left|+\right|{\hat{y}}_{i}|\right)}{2}}$$

Table 2 summarizes the performance metrics obtained for each model–dataset pair. To keep the results concise, only the models trained on the full, combined dataset are evaluated, as this variation in the training data generally leads to better performance measures. For validation against the Rhode Island scene, validation results are only reported for the OLCI, MODIS, and VIIRS sensors, as insufficient coincident data were available to evaluate the PlanetScope model. As the HICO evaluation scores are across three scenes, the metrics reflect the average score and runtime across all HICO scenes.

It is notable that the RMSE changes very little, while the $R^{2}$ value is slightly more sensitve and the SMAPE experiences large changes. This is the because the majority of the $R_{rs}$ values are very small decimal values. When such low-magnitude values dominate the dataset, relative error measures like $R^{2}$ and SMAPE provide more meaningful insight into the model’s performance, as they account for proportional differences rather than absolute ones. While the metrics of $R^{2}$ and SMAPE are generally stable on the in situ and HICO data, with $R^{2}$ increasing with the number of bands and SMAPE decreasing as the bands increase, interpreting the evaluation on the Rhode Island scene presents unique challenges. This is because the data is sourced from different sensors each passing at a different time of day and with their own resolution. Because of these differences, it can be expected that the features may vary slightly as well as the number of contaminated pixels that are removed from the scene, which will slightly impact the perceived performance. The PACE OCI run times are also skewed by the data resolution, as the OLCI data has a resolution of 300 m, the VIIRS has a resolution of 750 m, and the MODIS has a resolution of 250 m. This means that the MODIS model, while typically faster than the OLCI model, will take more time because it is running on more data points. The time to predict can vary greatly due to the increase in the number of data points predicted, but the performance measures remain fairly consistent, with the 7- and 11-band models achieving an $R^{2}$ nearly above 0.96 and SMAPE near or below 15 %; the 5-band model is also in this range for $R^{2}$ but lags slightly behind for SMAPE at nearly 18%, and the 3-band model struggles a bit more, with an $R^{2}$ between 0.87 and 0.95 and SMAPE reaching as high as 20%. This ability to stay in a close range despite increased data variability, source, and size is a good indicator that the true performance of these models is being captured. Overall, the results highlight variation in model accuracy across datasets, reflecting differences in data distribution and coverage. The following section provides a more detailed discussion of the performance trends.

#### 3.2.1. NOAA Cal/Val In Situ Data

The overall aim of the annual NOAA calibration and validation cruises is to support improvements in the extent and accuracy of satellite remotely sensed ocean color parameters in the near surface ocean by collecting high quality in situ optical and biogeochemical data for validating satellite ocean color radiometry and derived products from NOAA JPSS VIIRS and additional Navy operational ocean color satellites. The in situ data collected during the NOAA calibration and validation cruises used in this study were collected using the Spectral Evolution Incorporated’s PSR-1100F hyperspectral spectroradiometer on the bow of NOAA’s Research Vessels. The cruises selected for examination were the 2019 cruise off the United States East Coast and the 2022 cruise in Hawaii. The New England cruise ranged from 8–17 September 2019, which consisted of a total of 26 stations, and is representative of coastal and shelf waters. The Hawaii cruise ranged from 7–18 March 2022 with 16 station locations and captures a more stable blue water environment near the ocean color satellite standard calibration and validation site MOBY. The stations for each cruise are shown in Figure 3.

Radiometric data collected from each cruise were processed, yielding $R_{rs}$ just above the ocean surface. In total, this in situ cruise data collection consists of 44 hyperspectral measurements. To evaluate on this dataset, the performance measures were computed for the entire dataset, leading to the values given in Table 2. Overall on the NOAA Cal/Val cruise data, the predictions improved with an increase in bands, with the 11-band OLCI model achieving an $R^{2}$ of 0.9815 and SMAPE of 5.25 and the 3-band PlanetScope having an $R^{2}$ of 0.8797 and much larger SMAPE of 20.24. Breaking down the performance metrics by cruise demonstrated that the models achieved better metrics on the East Coast data than Hawaii. The 11-band OLCI model had an RMSE of 0.0001 for the East Coast data and Hawaii data and an SMAPE of 4.41 for the East Coast and 6.71 for Hawaii. The comparison was similar for the 3-band PlanetScope model, which had an RMSE of 0.0003 for the East Coast data and 0.0007 for Hawaii and an SMAPE of 17.03 for the East Coast and 25.86 for Hawaii. These performance differences can be partially attributed to the distinct seasonal and hydrological contexts of the two cruises. The East Coast cruise captured higher optical variability, enhancing the model’s ability to detect spectral patterns. In contrast, the Hawaii cruise, conducted in stable blue water conditions, exhibited nearly constant reflectance values above 600 nm. This lack of spectral contrast limited the models’ ability to differentiate features and, in some cases, even resulted in negative $R^{2}$ values.

Figure 4 shows the $R_{rs}$ collected at three of the stations for each cruise to demonstrate the differences in the true hyperspectral $R_{rs}$ values at each point and the hyperspectral predictions made by the four models. As seen in this figure, the majority of the models fit closely to the true hyperspectral curve, with the PlanetScope 3-band model falling off trend below 500 nm due to its lack of spectral bands below 490 nm. In this figure, the Hawaii cruise samples appear to be near-perfect fits at wavlengths above 490 nm even for the PlanetScope model, while there is more variation in the East Coast predictions. In this case, the $R^{2}$ and SMAPE are impacted heavily due to the uniformity of the Hawaii dataset because a simple prediction of the mean value for all points would outperform the models’ estimates. Additionally, the RMSE is impacted by the scale of the data since the East Coast data has a maximum value of 0.003 $sr^{-1}$, while the Hawaii data ranges as high as 0.0175 $sr^{-1}$, meaning the Hawaii data can have larger estimates in error simply due to the scale of the data. This highlights the limitations of relying solely on the performance measures of individual datasets to assess model performance under different environmental conditions. Examining performance of the models in a variety of oceanic contexts is important, as the ocean presents a wide range of optical properties each of which leads to a different spectral response.

#### 3.2.2. HICO Data

Next the models were applied to real, hyperspectral satellite datasets that were not included in the training and testing splits. The first of these is the data from HICO. The HICO sensor is a hyperspectral sensor that has a 5.7 nm spectral resolution. We selected three HICO scenes that were atmospherically corrected and exhibited a variety of water color. The HICO scenes presented here are mostly free from cloud contamination. For these scenes, the level 2 (atmospherically corrected) true color representation of these three scenes is shown in Figure 5. These true color products are generated using the surface $R_{rs}$ nearest to the 443, 551, and 671 nm wavelengths.

The HICO scenes were sized at (512, 2000). Land and cloudy pixels were omitted from the evaluation, leading to 278,269 pixels to predict for Vermillion Bay, 290,107 for Saint Joseph Bay, and 422,440 for the Key West scene. The models showed a general trend of predicting each scene in roughly 5 min, with time increasing slightly from the 3-band PlanetScope model to the 11-band OLCI model as shown in Table 2. The metrics achieved on each of the scenes, given in Table 3, reflect an overall increase in accuracy with the increase in bands and appear to remain fairly consistent despite the changes in ocean-color distributions across the scenes. Most of the models achieve an $R^{2}$ of over 0.99, with the exception being the PlanetScope 3-band model achieving closer to 0.95. Overall, the fit of each hyperspectral curve shown in Figure 6 also reflects this increase in accuracy for the models with increased bands, with most of PlanetScope’s inaccuracy again occurring below the 490 nm band.

#### 3.2.3. Rhode Island Data

To test the applicability of the hyperspectralization approach on raw sensor data, a scene off the coast of Rhode Island was selected, with coincident data gathered from MODIS, SNPP VIIRS, OLCI 3A, and PACE OCI. Despite differences in overpass times, spatial resolutions, and levels of cloud and sun glint contamination, the hyperspectralization models performed well across all sensors. Specifically, the Rhode Island scene was captured at 14:53 UTC by OLCI, 16:16 UTC by SNPP VIIRS, 17:45 UTC by PACE OCI, and 18:00 UTC by MODIS. While the core scene features remained largely consistent during this window (as shown in Figure 7), data quality varied—MODIS, for instance, was significantly affected by sun glint later in the day, whereas OLCI and VIIRS had clearer observations closer to the PACE OCI overpass.

These conditions likely influenced performance metrics: VIIRS achieved the highest accuracy with an average $R^{2}$ of 0.9851, SMAPE of 18.20%, and RMSE of 0.0007, followed by MODIS ($R^{2}$ = 0.9774, SMAPE = 15.45%, RMSE = 0.0008) and OLCI ($R^{2}$ = 0.9594, SMAPE = 14.84%, RMSE = 0.0011). Differences in spatial resolution—MODIS at 250 m, OLCI at 300 m, and VIIRS at 750 m—also impacted processing time. Although MODIS typically runs faster, its higher resolution led to longer prediction times due to the greater number of data points.

While direct comparisons to PACE OCI are complicated by sensor differences and the use of PACE OCI data as hyperspectral “ground truth,” this experiment provides valuable insights into real-world applicability. All models achieved SMAPE values under 20%, suggesting robust and consistent performance across varying input conditions. These results indicate that hyperspectralization can effectively preserve critical scene features, even across sensors with different temporal and spatial characteristics.

## 4. Discussion

During training, the models consistently performed best when trained on the combined synthetic and PACE OCI dataset. This outcome is expected: although the addition of synthetic data did not substantially increase the size of the training set, it did expand the chromaticity space covered, offering a broader spectral context for the models to learn from. This broader coverage may explain observed differences in validation performance. As shown in Figure 8, the chromaticity space of the HICO dataset closely aligns with the training data distribution, with only a few points falling on the edge of the space covered by the training data. Nearly half of the in situ points, however, trail along the edge of the training data’s chromaticity space. This disparity likely contributes to the models’ improved performance on the HICO dataset relative to the in situ data. Additionally, the Rhode Island scene spans a relatively narrow chromaticity space, which is fully encompassed by the training data, further supporting the high model accuracy observed in those comparisons.

While the 5-band, 7-band, and 11-band models accurately reconstructed the hyperspectral curve (Figure 4), the 3-band PlanetScope model struggled, particularly below 490 nm. This limitation is expected, as the model lacks input data at lower wavelengths and must extrapolate beyond its available range. Interestingly, the 3-band model showed better predictions at lower wavelengths for the HICO dataset compared to the in situ dataset. This suggests that certain dataset characteristics, such as scene variability or alignment with training spectra, can mitigate the limitations of reduced spectral input. In contrast, the models with broader band coverage, particularly in the low 400 nm and high 600 nm ranges, consistently performed well. This highlights the importance of spectral diversity in both training data and input bands, especially when predicting outside the native sensor range.

A recurring feature in the model predictions is a dip in the hyperspectral curve near 690 nm, as seen in Figure 4 and Figure 6. This feature reflects a pattern present in the training data itself (Figure 9), likely introduced during atmospheric correction in the APS-processed PACE OCI data. While the presence of this dip in model outputs demonstrates their capacity to reproduce detailed spectral features seen during training, it also raises concerns about learning artifacts specific to the training pipeline. As a potential improvement, future work should investigate whether removing such artifacts prior to training can reduce overfitting to dataset-specific features and improve generalization to other hyperspectral datasets.

Despite these limitations, the hyperspectralizers produced realistic and accurate spectral reconstructions across a range of scenarios. High $R^{2}$ values and low error metrics across the NOAA in situ measurements, HICO scenes, and the Rhode Island test scenes support the effectiveness of the approach. In particular, the comparisons with PACE OCI highlight the models’ ability to recover $R_{rs}$ values at wavelengths between 400 and 700 nm with strong fidelity, even from multispectral inputs of varying spectral resolution and quality. These results underscore the robustness of the hyperspectralization framework and its potential for wide applicability in remote sensing tasks.

## 5. Conclusions

This study demonstrates the potential of using machine learning, specifically Gradient Boosting, to reconstruct hyperspectral $R_{rs}$ in the visible spectrum from limited MSI bands. In doing so, we bridge the gap between widely available, lower-cost multispectral data and the rich spectral information traditionally obtainable only through hyperspectral sensors. Our results show that models trained on a combined dataset of synthetic and PACE OCI-derived spectra can effectively predict hyperspectral signatures across diverse coastal ocean environments, achieving high fidelity even on unseen in situ and satellite data.

The analysis highlights key factors that influence model performance, including the diversity of the training dataset’s chromaticity space and the spectral coverage of the input bands. While broader-band models (e.g., 5-band, 7-band, and 11-band) generally outperform limited-band models (e.g., 3-band), results from the HICO dataset suggest that even minimal MSI inputs can provide meaningful spectral reconstructions when the model is trained on data representative of the target domain.

Overall, this approach offers a promising pathway for expanding the utility of MSI data, especially from UAVs and nanosatellite platforms, which often lack bands that are key in atmospheric correction but could now be used to provide high-resolution ocean color products. Future work will explore extending the supported sensors, further evaluating model generalizability, using the hyperspectralized data to create more complete global coverage, and using the extended bands to enhance ocean-color products. With continued development, spectral super-resolution could play a pivotal role in scalable, cost-effective coastal and marine monitoring.

## Figures and Tables

**Figure 1 sensors-25-06389-f001:**
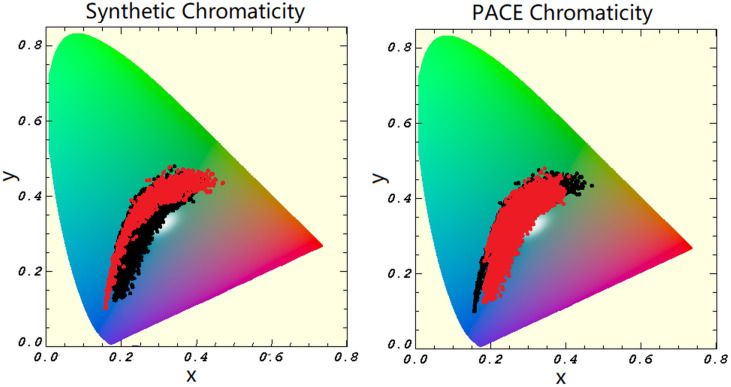
The chromaticity space covered by the synthetic dataset (**Left**), the global PACE OCI dataset (**Right**). Each dataset is shown in red with the combined data space shown in the background (black) for comparison.

**Figure 2 sensors-25-06389-f002:**
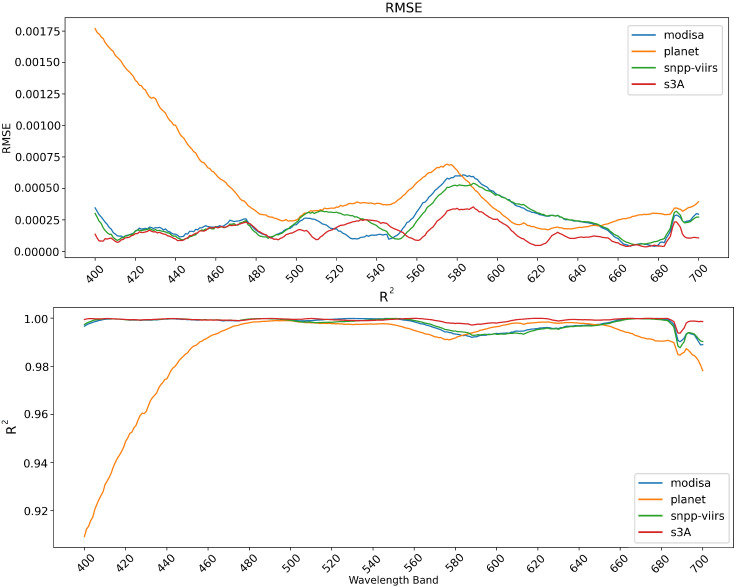
The $R^{2}$ and RMSE per wavelength band for each of the models (MODIS, VIIRS, OLCI, Planet) trained on the full dataset.

**Figure 3 sensors-25-06389-f003:**
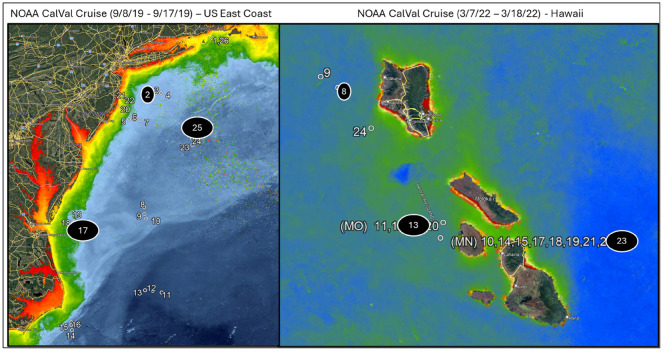
The stations for each of the NOAA Cal/Val cruises used in this evaluation. Stations 1–26 for the 2019 East Coast cruise are shown on the left and stations 8–23 for the 2022 Hawaii cruise are on the right.

**Figure 4 sensors-25-06389-f004:**
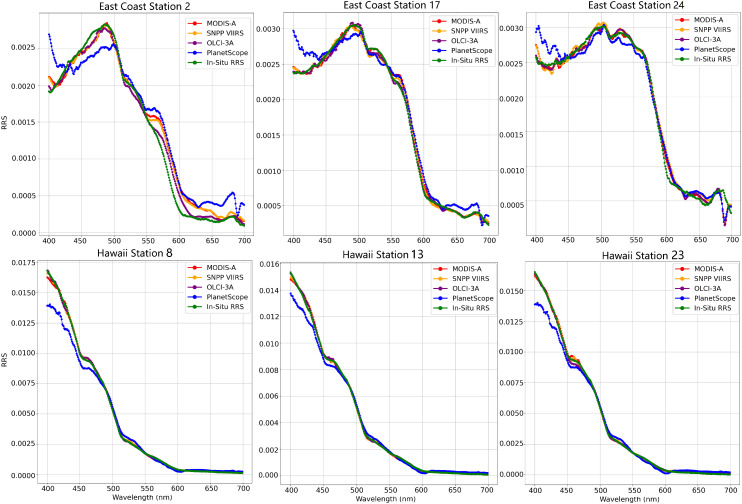
The performance of the estimated hyperspectral curve for each model on data from three randomly selected East Coast stations (**top**) and three randomly selected Hawaii stations (**bottom**), and the in situ $R_{rs}$ data is compared.

**Figure 5 sensors-25-06389-f005:**
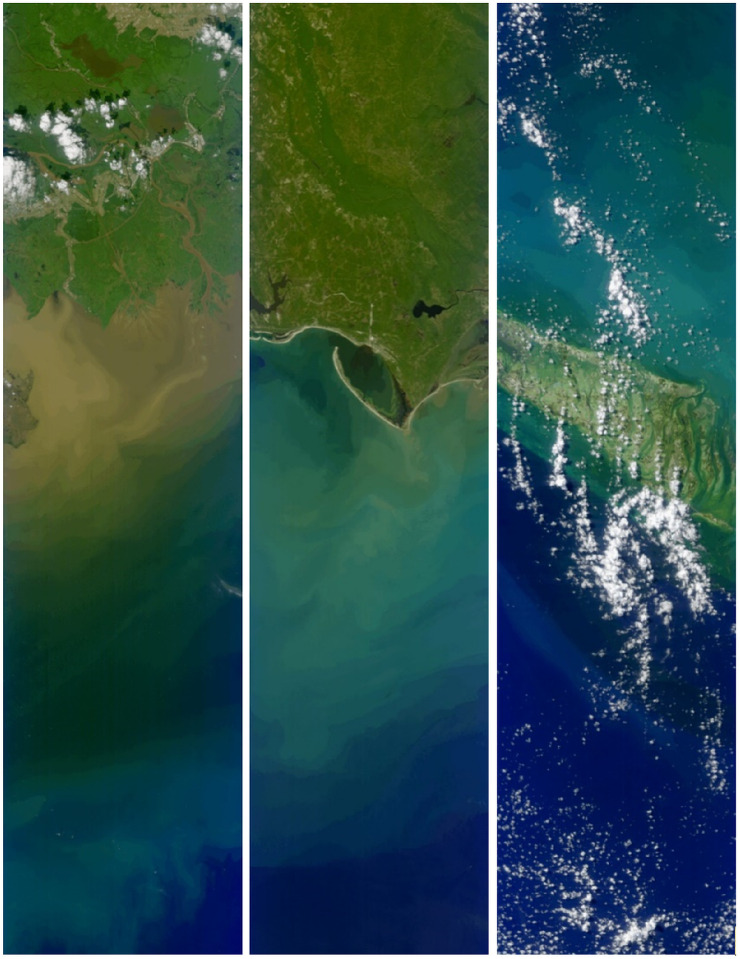
Three HICO scenes used for model-predicted $R_{rs}$ compared to APS-derived $R_{rs}$. Vermillion Bay, Lousiana (**Left**); Saint Joseph Bay, Florida (**Middle**); Key West, Florida (**Right**).

**Figure 6 sensors-25-06389-f006:**
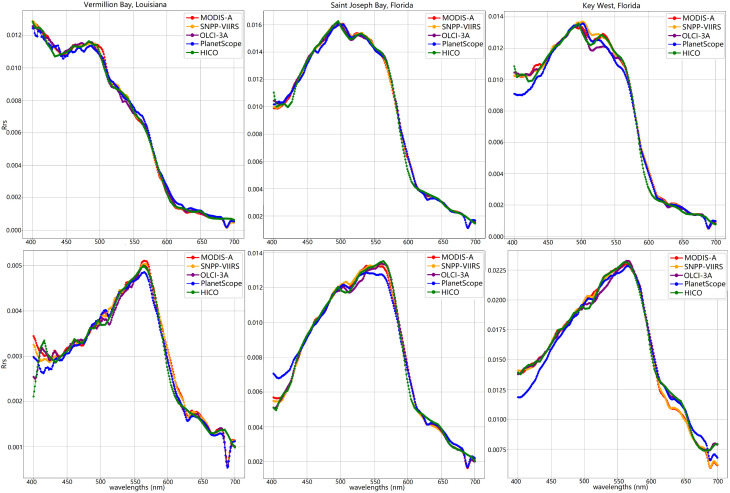
Two points—one near the bottom, one near the top of the valid data—were taken for each of the HICO scenes to compare the hyperspectral $R_{rs}$ predictions for Vermillion Bay (**Left**); Saint Joseph Bay (**Middle**); Key West_ISS.nc (**Right**).

**Figure 7 sensors-25-06389-f007:**
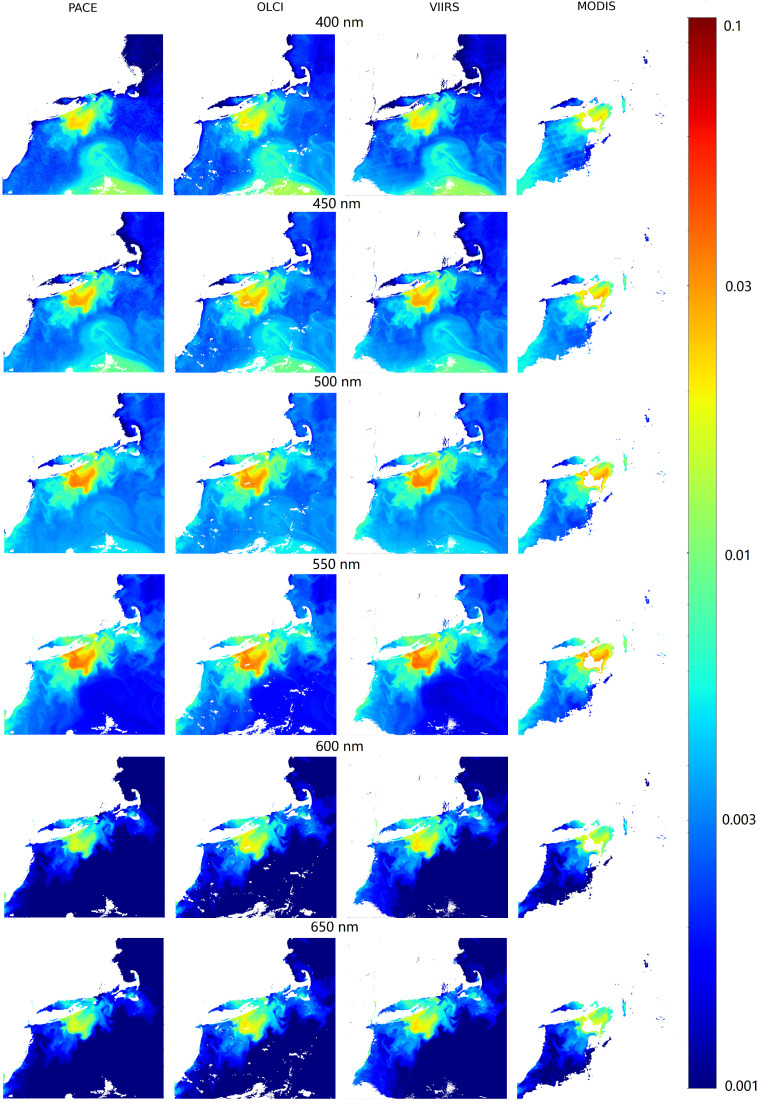
Hyperspectralized output of each sensor (MODIS, VIIRS, and OLCI) is compared to PACE OCI for wavelengths in 50 nm increments.

**Figure 8 sensors-25-06389-f008:**
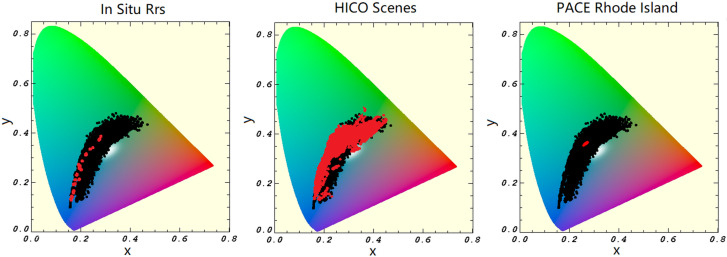
Here we compare the covered chromaticity space for the validation sets versus the training set. The in situ dataset is depicted to the right, HICO in the middle, then in the far right image is the PACE OCI Rhode Island scene. The chromaticity space of each validation dataset is displayed in red and overlaid on the full training data space (black).

**Figure 9 sensors-25-06389-f009:**
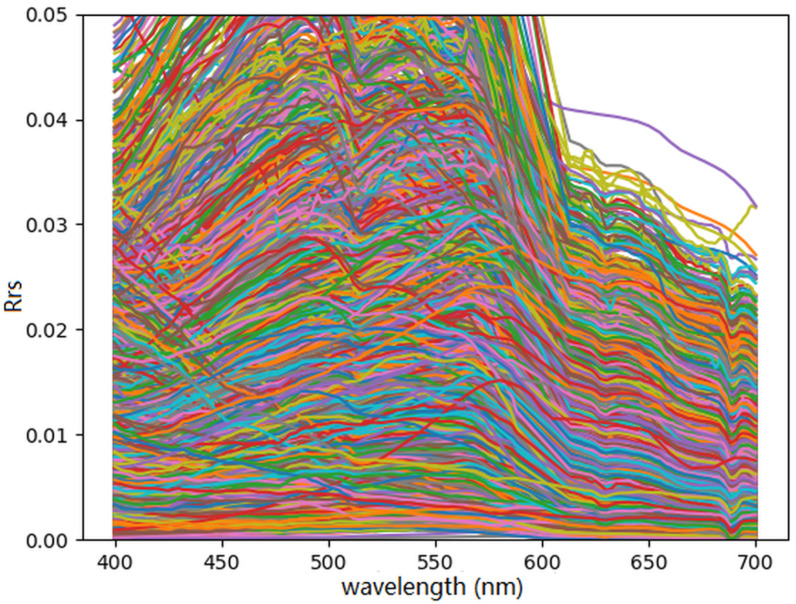
The spectra used in training for the combined synthetic and PACE OCI dataset was plotted as a collection of hyperspectral curves with varying colors to enhance visibility.

**Table 1 sensors-25-06389-t001:** Performance of each of the models trained on the synthetic dataset, the PACE OCI dataset, and the combined data is compared in terms of $R^{2}$ and RMSE obtained on the 20% validation split and the total time to train.

Model	$R^{2}$	RMSE (${\mathrm{sr}}^{-1}$)	Training Time
OLCI 11-band on synthetic data	0.99688	0.00021	2 min 32 s
OLCI 11-band on PACE OCI data	0.99915	0.00016	33 min 26 s
OLCI 11-band on all data	0.99919	0.00015	36 min 16 s
MODIS 7-band on synthetic data	0.99506	0.00026	1 min 49 s
MODIS 7-band on PACE OCI data	0.99801	0.00023	21 min 44 s
MODIS 7-band on all data	0.99770	0.00024	23 min 32 s
VIIRS 5-band on synthetic data	0.99443	0.00027	1 min 18 s
VIIRS 5-band on PACE OCI data	0.99790	0.00024	15 min 54 s
VIIRS 5-band on all data	0.99761	0.00024	17 min 5 s
PlanetScope 3-band on synthetic data	0.98577	0.00047	52 s
PlanetScope 3-band on PACE OCI data	0.98773	0.00053	9 min 58 s
PlanetScope 3-band on all data	0.98803	0.00052	10 min 32 s

**Table 2 sensors-25-06389-t002:** Performance of each of the models when run on the in situ $R_{rs}$, Rhode Island PACE OCI scene, and HICO data is compared in terms of $R^{2}$, SMAPE, RMSE, and time to predict.

Model	$R^{2}$	SMAPE (%)	RMSE (${\mathrm{sr}}^{-1}$)	Prediction Time
OLCI 11-band on in situ $R_{rs}$	0.9815	5.25	0.0001	1.16 s
OLCI 11-band on RI Scene	0.9594	14.84	0.0011	21 min 36 s
OLCI 11-band on HICO Data	0.9943	4.48	0.0002	6 min 30 s
MODIS 7-band on in situ $R_{rs}$	0.9634	8.37	0.0001	0.86 s
MODIS 7-band on RI Scene	0.9774	15.45	0.0008	67 min 26 s
MODIS 7-band on HICO Data	0.9913	5.98	0.0004	6 min 23 s
VIIRS 5-band on in situ $R_{rs}$	0.9510	9.51	0.0001	0.96 s
VIIRS 5-band on RI Scene	0.9851	18.20	0.0007	12 min 19 s
VIIRS 5-band on HICO Data	0.9911	6.35	0.0003	6 min 19 s
PlanetScope 3-band on in situ $R_{rs}$	0.8797	20.24	0.0005	0.97 s
PlanetScope 3-band on HICO Data	0.9518	9.68	0.0008	6 min 16 s

**Table 3 sensors-25-06389-t003:** Performance of each model when run on different regions (Vermillion Bay, Saint Joseph Bay, Key West), compared in terms of $R^{2}$, SMAPE, RMSE, and prediction time.

Region and Model	$R^{2}$	SMAPE (%)	RMSE (${\mathrm{sr}}^{-1}$)	Prediction Time
Vermillion Bay				
OLCI 11-band	0.9958	4.64	0.0002	5 min 37 s
MODIS 7-band	0.9925	5.69	0.0004	5 min 32 s
VIIRS 5-band	0.9923	5.86	0.0003	5 min 29 s
PlanetScope 3-band	0.9473	10.41	0.0009	5 min 22 s
Saint Joseph Bay				
OLCI 11-band	0.9915	3.75	0.0003	5 min 32 s
MODIS 7-band	0.9893	4.70	0.0003	5 min 34 s
VIIRS 5-band	0.9892	5.14	0.0003	5 min 29 s
PlanetScope 3-band	0.9479	8.16	0.0007	5 min 28 s
Key West				
OLCI 11-band	0.9955	5.04	0.0003	8 min 21 s
MODIS 7-band	0.9920	7.54	0.0004	8 min 2 s
VIIRS 5-band	0.9918	8.05	0.0004	7 min 59 s
PlanetScope 3-band	0.9602	10.46	0.0009	7 min 58 s

## Data Availability

The synthetic dataset used in training is available at https://ioccg.org/resources/data, accessed on 15 August 2025. The HICO data is available for download at https://oceancolor.gsfc.nasa.gov, accessed on 15 August 2025. The Rhode Island satellite data and PACE training data are available on request.

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
