# Peer review of "Gradient Boosting for the Spectral Super-Resolution of Ocean Color Sensor Data"

_sensors, 2025, doi:10.3390/s25206389_

Round 1

Reviewer 1 Report

Comments and Suggestions for Authors

The manuscript presents a machine learning framework using gradient boosting to reconstruct hyperspectral ocean color data (400–700 nm) from multispectral sensor inputs. The study is timely and relevant given the increasing availability of low-cost multispectral platforms and the high demand for hyperspectral-like products in ocean color remote sensing. The methodology is clearly explained, results are thorough, and validation against in situ, satellite (HICO), and coincident multispectral-hyperspectral data strengthens the contribution.

Major Comment 1:
The introduction cites SSR and deep learning literature but does not clearly contrast gradient boosting with other recent approaches (e.g., CNNs, transformers, unfolding-based networks). I suggest explicitly positioning gradient boosting relative to deep learning.

Major Comment 2:

The paper states that a gradient boosting algorithm was used but does not specify the implementation (e.g., XGBoost, LightGBM, Scikit-learn) or the key hyperparameters (e.g., learning rate, number of trees, tree depth). A brief subsection or a reference to a table in supplementary materials detailing the model configuration is necessary for reproducibility.

Minor comments:
1. Line 21: Suggest using "in-situ" to avoid any confusion.

2. Line 136: Should be "Gradient Boosting was chosen".

3. Equation 5: Should this be the (y(i,j)-y_hat(i,j))^2?

4. Line 248: The time period is not consistent with Figure 3 (March 7-18).

5. Line 348: Should this be Figures 4 and 6?

6. Please ensure all axes in the figures have units as well as the table captions (RMSE, SMAPE).

Author Response

Thank you for the detailed and constructive feedback. The following comments were addressed and will be highlighted in the fixed revision.

Major Comment 1:
The introduction cites SSR and deep learning literature but does not clearly contrast gradient boosting with other recent approaches (e.g., CNNs, transformers, unfolding-based networks). I suggest explicitly positioning gradient boosting relative to deep learning.

We appreciate this feedback. A paragraph has been added to address the choice of a smaller, gradient boosting framework as opposed to a deep learning approach.

Major Comment 2:

The paper states that a gradient boosting algorithm was used but does not specify the implementation (e.g., XGBoost, LightGBM, Scikit-learn) or the key hyperparameters (e.g., learning rate, number of trees, tree depth). A brief subsection or a reference to a table in supplementary materials detailing the model configuration is necessary for reproducibility.

This was absolutely an oversight. This key information has been added to the materials and methods section.

Minor comments:
1. Line 21: Suggest using "in-situ" to avoid any confusion. Fixed

  1. Line 136: Should be "Gradient Boosting was chosen". Fixed
  2. Equation 5: Should this be the (y(i,j)-y_hat(i,j))^2? Good catch. Fixed.
  3. Line 248: The time period is not consistent with Figure 3 (March 7-18). Addressed.
  4. Line 348: Should this be Figures 4 and 6? You are correct.
  5. Please ensure all axes in the figures have units as well as the table captions (RMSE, SMAPE). Units have been added.

Reviewer 2 Report

Comments and Suggestions for Authors

The reviewed manuscript is devoted to new technical possibilities for studying ocean color, which is of both technical and scientific importance. The work describes the methodology and results of the investigation, accompanied by figures that meet the publisher's quality requirements, and is written in good scientific and technical language. In my opinion, after minor revisions, the manuscript can be approved for publication. To this purpose, I would recommend that the authors pay attention to the following points.

  1. In the introduction, the authors list the satellite sensors used to obtain hyperspectral data. They state that “...pathways for utilizing these high-dimensional datasets are in their early stages” (lines 23-24). It would be useful to provide at least a brief description of these sensors, noting their strengths and weaknesses. This is particularly important because data from PACE OCI was used in the processing of high-resolution hyperspectral signatures (Materials and Methods section).
  2. Lines 64-66: “...wide range of different water turbidities using variable Inherent Optical Properties (IOPs) as input to the HYDROLIGHT numerical radiative transfer simulation.” Explain how (on what basis) and in what classification water turbidity was determined. What do IOPs include? Provide a description of the HYDROLIGHT numerical radiative transfer simulation.
  3. What is the principle of the Relative Spectral Response (RSR) function?
  4. In the section on colorimetry, the authors refer to rather old publications. Taking into account the importance of the works mentioned and without doubting their high quality, it would be advisable to provide a more up-to-date overview of this topic.
  5. In my opinion, Figure 1 does not clearly show the differences between the synthetic dataset, the global PACE OCI dataset, and the combined chromatic space.
  6. I would recommend providing a more technically complete description of the gradient boosting algorithm. This is important because in this particular case, the authors apply machine learning techniques, which is one of the stated topics of the work.
  7. The simulation results presented in the tables and described in the text require more in-depth analysis. For example, why does the R2 parameter remain essentially unchanged despite significant variation in Prediction and Training Time?
  8. To examine the results, the authors used data from two cruises, in 2019 (off the United States East Coast) and 2022 (Hawaii). The studies were conducted over short periods of time in different months of the year and under different hydrological conditions. I would like to see an analysis of the impact of these differences on the modeling results in the manuscript.
  9. In conclusion, I would like to note once again that, overall, the work is undoubtedly interesting, and I hope that the authors will not find it difficult to take these recommendations into account.

Author Response

  1. In the introduction, the authors list the satellite sensors used to obtain hyperspectral data. They state that “...pathways for utilizing these high-dimensional datasets are in their early stages” (lines 23-24). It would be useful to provide at least a brief description of these sensors, noting their strengths and weaknesses. This is particularly important because data from PACE OCI was used in the processing of high-resolution hyperspectral signatures (Materials and Methods section). A brief description of the two sensors has been added to this paragraph.
  2. Lines 64-66: “...wide range of different water turbidities using variable Inherent Optical Properties (IOPs) as input to the HYDROLIGHT numerical radiative transfer simulation.” Explain how (on what basis) and in what classification water turbidity was determined. What do IOPs include? Provide a description of the HYDROLIGHT numerical radiative transfer simulation. The hydrolight IOP considerations and radiative transfer explanation has been expanded.
  3. What is the principle of the Relative Spectral Response (RSR) function? This information has been added to the Convolutions section
  4. In the section on colorimetry, the authors refer to rather old publications. Taking into account the importance of the works mentioned and without doubting their high quality, it would be advisable to provide a more up-to-date overview of this topic. We agree on the importance of referencing foundational work in colorimetry. Colorimetric principles, including the CIE 1931 color space and ΔE metrics, have been well-established for decades and continue to underpin both theoretical and applied color science. Given that the physical and perceptual basis of colorimetry has remained consistent, our citations focused on these works. That said, in response to the reviewer’s suggestion, we have updated the manuscript to include several more recent references that highlight ongoing developments in the application and refinement of colorimetric techniques
  5. In my opinion, Figure 1 does not clearly show the differences between the synthetic dataset, the global PACE OCI dataset, and the combined chromatic space. The figure has been adjusted to help make this more clear
  6. I would recommend providing a more technically complete description of the gradient boosting algorithm. This is important because in this particular case, the authors apply machine learning techniques, which is one of the stated topics of the work. The description of the gradient boosting regressor has been expanded per this and other reviewers’ requests.
  7. The simulation results presented in the tables and described in the text require more in-depth analysis. For example, why does the R2 parameter remain essentially unchanged despite significant variation in Prediction and Training Time? We have added more analysis on the metrics.
  8. To examine the results, the authors used data from two cruises, in 2019 (off the United States East Coast) and 2022 (Hawaii). The studies were conducted over short periods of time in different months of the year and under different hydrological conditions. I would like to see an analysis of the impact of these differences on the modeling results in the manuscript. We have added more a more detailed explanation of the performance differences and how the conditions impact this.
  9. In conclusion, I would like to note once again that, overall, the work is undoubtedly interesting, and I hope that the authors will not find it difficult to take these recommendations into account. Thank you so much for the kind feedback. All of your suggestions have been taken into account.